# The Effect of the Cooling Process on the Crystalline Morphology and Dielectric Properties of Polythene

**DOI:** 10.3390/ma13122791

**Published:** 2020-06-20

**Authors:** Guang Yu, Boyang Yu

**Affiliations:** 1Mechanical and Electrical Engineering Institute, University of Electronic Science and Technology of China, Zhongshan Institute, Zhongshan 528400, China; 2Mechanical and Electrical Engineering Institute, Zhongshan Polytechnic, Zhongshan 528400, China; 0201965001@zspt.cn

**Keywords:** LDPE, cooling process, crystalline morphology, dielectric properties

## Abstract

In this study, LDPE samples were prepared by melt blending with different cooling processes, which were natural air cooling, rapid air cooling, water cooling and oil cooling, respectively. According to polarization microscope (PLM) and differential scanning calorimeter (DSC) tests of these four low-density polyethylene (LDPE) samples, the effect of different cooling processes on polythene crystalline morphology could be studied. According to conductivity, dielectric frequency spectra and space charge tests, the effect of crystalline morphology on dielectric macroscopic properties could be explored. The microstructure characteristic results indicated the cooling medium significantly affected polythene crystalline morphology. When the samples were produced with natural air cooling, the crystalline grain size was large. On the other hand, after rapid air cooling, water cooling and oil cooling processes, the samples’ crystalline grain dispersed uniformly, and the grain sizes were lower. The space charge testing results indicate the samples produced with water cooling and oil cooling processes restrained the electrode injection in the process of pressurization. During short-circuits, the rates of charge release of these two samples were fast, and the remaining space charges were fewer. The conductivity and dielectric frequency spectra testing results indicated the conductivities of samples produced with water cooling and oil cooling processes were both less than those of samples produced with a natural air cooling process. Besides, with increasing experimental frequency, the relative dielectric constants of all testing samples decreased. Among them, the relative dielectric constant of the LDPE sample with the natural air cooling process was the largest. However, the crystalline structures of samples produced with rapid air cooling and water cooling processes were close, which restrained the movement of polymer macromolecule chains. Thus, the dielectric constants were lower. Additionally, because of the influence of relaxation polarization and dipole polarization, the dielectric losses of LDPE with water cooling and oil cooling processes increased to varying degrees.

## 1. Introduction

With rapid economic development, there has been an ever-growing demand for electrical energy. New types of power transmission technology, such as high-voltage direct current transmission (HVDCT) and flexible AC transmission (FACT), have emerged. On the other hand, insulation materials’ aging caused by the space charge under DC high voltage (DCHV) is serious [1,2,3,4]. Because of their excellent electrical insulating properties and processing properties, polymer materials are widely used in the electrical and electronic insulation fields. The application of polymer blend technology has made great contributions to the development of this new type of insulating material. Especially, nanoparticles possess unique physical-chemical properties, such as large specific surface area and high surface activity. A small amount of nanoparticle doping can obviously improve polymer-based composites’ properties. Some researchers found that crystalline morphology was one of the main factors which affected the dielectric properties of polymer composites, such as breakdown property, conductivity and space charge [5,6,7,8,9,10]. In order to ensure the transfer capability and operational stability of a power system, the dielectric properties of insulating materials were improved by a crystalline structure change in the polymer composites [11,12,13,14]. In this research, the crystalline structure of polyethylene cables was greatly affected by cooling methods. The different microstructures then affect the macroscopic dielectric properties of the insulating materials. For example, both crystal size and crystalline integrity affect composites’ conductivity characteristics [15,16,17,18,19]. Space charge generation, movement and attenuation are not only linked to the applied voltage, but also to the polymer crystalline state [20,21,22]. Therefore, the effect of different cooling processes on polymers’ crystalline structure deserves to be studied. Additionally, this will provide a theoretical foundation for follow-up research on the effects of crystalline morphology and interfacial structure on composites’ macroscopic properties.

In this article, low-density polythene (LDPE) insulation material was prepared by melt blending and LDPE sheeting samples were pressed by a plate vulcanizing press machine using different methods. These were natural air cooling, rapid air cooling, water cooling and oil cooling. According to the PLM and DSC tests, the samples’ crystalline morphology was observed, from which the effects of different cooling media on materials’ crystalline morphology were analyzed. According to pulsed electroacoustics, the space charge accumulation and attenuation of each sample under pressure and their short-circuit conditions could be measured, from which the different cooling medias’ effects on polythene space charge behavior could be explored. The volt-ampere characteristic testing equipment was built based on picoammeters. Thus, the relationship between samples’ conductivity and field strength was measured, from which the effects of different cooling media on materials’ conductivity were analyzed. The dielectric spectra of polythene and its nanocomposites were also measured. The dielectric constant and loss factor change with frequency were recorded, from which the effects of different cooling media on the dielectric constant and loss factor could be explored.

## 2. Experimental Methods and Sample Preparation

### 2.1. Sample Preparation

In sample preparation, firstly, the LDPE particles were added into the torque rheometer, the temperature of which was set to 150 °C. When the particles achieved the melting state, these samples were pressed into sheets by plate vulcanizing press machine (The temperature was 150 °C, the pressure was 15 MPa and the duration was 15 min). In order to explore the effects of different cooling media on the crystalline morphology and dielectric properties of LDPE, these samples were cooled down to room temperature with different cooling media. The samples’ thicknesses were 100 μm, 200 μm and 300 μm respectively, and all of them were dealt with PLM test, DSC test, dielectric frequency spectra test, conductivity test and space charges test.

In the cooling process, the LDPE samples were cooled by the processes of natural air cooling, rapid air cooling, water cooling and oil cooling respectively. The specific implementation process was as follows:Natural air cooling: After melt pressing for 15 min with a 150 °C plate vulcanizing press machine, the samples were taken out and placed into another plate vulcanizing press machine at room temperature. Then, the 10 MPa pressure was maintained until the samples were cooled down to room temperature.Rapid air cooling: After melt pressing, the samples were taken out of the 150 °C plate vulcanizing press machine and exposed to the air. Then the samples were processed with rapid cooling by an electric fan.Water cooling: After melt pressing, the samples were taken out of the 150 °C plate vulcanizing press machine and put into warm water rapidly for cooling. All these samples were then placed in air and dried.Oil cooling: After melt pressing, the samples were taken out of the 150 °C plate vulcanizing press machine and put into cable oil rapidly for cooling. All these samples were then placed in air and dried.

After the cooling process, all the samples were place in a drying tank for the relevant test. The samples prepared with natural air cooling, rapid air cooling, water cooling and oil cooling were marked A1, A2, W3 and O4.

### 2.2. Structure Characterization and Performance Test of Different Samples

According to the PLM test, the crystalline morphology and crystal sizes of different samples could be observed. The test samples’ thickness was 100 μm. These samples surfaces were corroded with a blend of potassium permanganate and concentrated sulfuric acid. After ultrasonic cleaning, the samples were placed on a glass slide and LDPE crystallization was observed by PLM.

The isothermal crystallization and melting process of the LDPE polymer were observed by a DSC test. The test samples’ thickness was 100 μm. Under 150 mL/min drying nitrogen, the samples were heated to 140 °C at a uniform speed of 10 °C·min^−1^. The enthalpy change during the heating process was recorded, from which the materials melting peak temperature *T*_m_ and melting enthalpy Δ*H*_m_ were obtained. These samples were then cooled down to 25 °C at the same speed. According to the enthalpy change in the cooling process, the crystallization peak temperature *T*_c_ and half wide of exothermic crystallization peak Δ*T*_c_ were obtained. Combining the parameters above and Equation (1), the different samples’ crystallinities were calculated.
(1)$$X_{c}=\frac{\Delta H_{m}}{(1-\omega )H_{0}}\times 100\%$$

In the conductivity test, the different samples’ conductivities were tested with a picoammeter. The test samples’ thickness was 200 μm. Ten measurements per sample were made, then the results were averaged. The boost method was step-up. The aluminum three-electrode system, of which the diameter was 50 mm, was evaporated on the samples’ surface by a vacuum coating machine. The samples’ conductivity could be calculated by Equation (2) [23].
(2)$$\sigma =\frac{I}{U}\times \frac{4d}{\pi (D_{1}\times g)}$$

In Equation (2), *σ* is the conductivity; *U* is the test voltage; *d* is the samples’ thickness; *D*_1_ is the inner electrode diameter on the samples’ upper surface; and *g* is the gap distance, which was 1 mm.

In this article, the broadband dielectric spectra analyzer was used for the dielectric frequency spectra test. The test samples’ thickness and diameter were 100 μm and 40 mm, respectively, and the experimental frequency range was 1–10^5^ Hz. The aluminum electrode system, of which the diameter was 25 mm, was evaporated on the samples’ upper and lower surface by a vacuum coating machine. In order to eliminate the effect of moisture and residual charges, the samples were placed in an 80 °C oven and treated with a short circuit for 24 h. Under room temperature, the relative dielectric constant and the loss factor of different samples were measured.

Pulsed electroacoustics was used for the space charge distribution test. The test samples’ thickness and diameter were 300 μm and 80 mm, respectively. Under the 10 kV/mm, 20 kV/mm and 40 kV/mm applied field strengths for 20 min pressure and short-circuit treatment for 30 min, the space charge distributions of the different samples were observed.

## 3. Experimental Results and Analysis

### 3.1. PLM Test Results and Analysis of Different Samples

The PLM patterns of samples prepared with different cooling mediums are shown in Figure 1. The polymer crystal possessed birefringence characteristics, which led to polarized light interference. There were light and dark areas in the crystalline patterns, and the light areas were polymer crystal.

The crystalline morphology of sample A1 is shown in Figure 1a. The grain sizes within the samples were larger, and the grains dispersed non-uniformly. Additionally, larger amorphous regions existed between the grains. This is because the cooling rate of natural air cooling was lower and crystal growth time was longer. Therefore, the nucleation number was lower and the grain sizes were larger. The crystalline morphology of sample A2 is shown in Figure 1b. Compared with Figure 1a, the grain size decreased significantly, and the grains dispersed uniformly. This is because the cooling rate of sample A2 was higher and crystal growth time was shorter. Thus, the nucleation number was larger and the grain size decreased. The crystalline morphology of samples W3 and O4 is shown in Figure 1c,d, respectively. The grain sizes were very small, and the grain distribution was more uniform. This illustrated that the cooling medium affected polythene crystalline morphology greatly. The relevant results also showed that the grain sizes of natural cooling samples were large, as opposed to the rapid cooling samples. This is consistent with the PLM test results [24].

### 3.2. DSC Test Results and Analysis of Different Samples

The heating and cooling process curves of different samples are shown in Figure 2. From the heating curve, the samples’ melting enthalpy Δ*H*_m_ and melting peak temperature were obtained. From the cooling curve, the crystallization peak temperature *T*_c_ and polymer crystallization rate Δ*T*_c_ were obtained. The isothermal crystallization and melting process parameters of the different samples are shown in Table 1.

Combining the test results for Figure 2 and Table 1, the melting temperatures, *T*_m_, of samples A2, W3 and O4 were all lower than that of sample A1. On the other hand, the crystallization peak of sample A1 was the lowest. This illustrates that the samples W3 and O4 crystallized at a high temperature. Among these, the crystallization peak *T*_c_ of sample O4 was the highest, and the half-wide exothermic crystallization peak Δ*T*_c_ was the lowest. Δ*T*_c_ was the polymer crystallization rate, so sample O4 possessed a fast crystallization rate, high crystallization temperature and perfect crystalline morphology. The samples W3 and A2 took second place. According to the DSC test results, the crystalline morphology and crystal sizes of the four samples in PLM patterns were verified further. The different cooling methods and media may have affected the polymers’ crystalline morphology.

### 3.3. Conductivity Characteristics Test Results and Analysis of Different Samples

The curves of different LDPE samples’ conductivity changing and field strength are shown in Figure 3.

From Figure 3, LDPE conductivity under high and low field strengths showed different change trends. In order to show the conductivity changes in LDPE samples with different cooling processes under high and low field strengths more clearly, the conductivity changes under low field strengths are listed in Figure 3 separately. Under low field strengths (lower than 10 kV/mm), the conductivity was relatively low. With the field strength increasing, the samples’ conductivity changed slightly. Under high field strength (higher than 10 kV/mm), the conductivity increased with field strength. This is because crystalline regions and amorphous regions coexisted in the polythene, and the traps existed between the crystalline regions and the amorphous regions. The trap structure had some effect on samples’ conductivity, which was the space charge limiting current effect [25]. Under low field strength, the traps were not filled up by electric charges. The carrier moving in the sample could be captured by the traps, which reduced the carrier concentration. Therefore, the conductivity was relatively low. With increasing field strength, the quantity of charge injected by the electrode increased significantly. When the traps were filled up by electric charges, the carrier was no longer captured. With increasing field strength, the carrier concentration increased greatly, and the samples’ conductivity was higher.

The different cooling mediums had some effect on LDPE conductivity, and the conductivities under high field strength and low field strength were different. In order to obtain a better comparison, conductivity line charts and bar charts for different cooling samples under 10 kV/mm and 50 kV/mm electric fields were drawn, which are shown in Figure 4. Under a 10 kV/mm electric field, the conductivity of different cooling samples differed widely. The conductivity of sample A1 was the least, while the conductivity of sample O4 was the greatest. Under a 50 kV/mm electric field, there was little difference between different cooling samples’ conductivities. The conductivities of samples A1 and A2 were greater, and the conductivities of samples W3 and O4 were lower.

This was because the electrodes could not inject charges into the LDPE under a low electric field. The ions played a key role in electric conduction, so it was ionic conduction. In the preparation of different samples, more impurity ions were introduced, which increased the carriers’ concentration. Therefore, the conductivity also increased. Under a high electric field, the electrodes injected more electrons and holes. The electrons, holes and impurity ions all participated in electric conduction. Among them, the electrons played the key role, so it was electronic conduction [26]. With a large number of charges injected by the electrodes, the carrier concentration increased. Therefore, there was little difference between different cooling samples’ conductivities. The conductivities of samples W3 and O4 were lower. This was because of the action of the cooling agent, which changed the crystalline morphology of LDPE. The grain sizes decreased. The traps between crystalline areas and non-crystalline areas increased, and more carriers were captured. Therefore, the conductivity and carrier mobility decreased.

### 3.4. Test Results and Analysis of Dielectric Frequency Spectra Characteristics

The different cooling processes changed the polythene microstructure, which also affected the polythene polarization. At the same time, the samples’ dielectric constants and loss were changed. Therefore, the effect of the cooling process on dielectric constant and dissipation factor of LDPE could be explored by a dielectric frequency spectra test. The changing relationships between the dielectric constant and dielectric loss for different samples with frequency are shown in Figure 5 and Figure 6, respectively.

From Figure 5, with the increase in frequency, the relative dielectric constants of the different samples all decreased at different degrees. This is because, at a low frequency (1 Hz), the polarization could be established completely, and the relative dielectric constants were fairly large. Meanwhile, at a high frequency (10^5^ Hz), a small part of polarization could not keep up with the applied electric field change, so polarization could not be established. In addition, the relative dielectric constant decreased slightly [27,28]. Among the samples, the relative dielectric constant of A1 was the largest, but the relative dielectric constants of samples A2 and W3 were relatively small. This is because the different cooling medium changed the crystalline morphology of polythene in various degrees. The grain sizes decreased, and the crystal structure was close, which limited the movement of the LDPE molecule chain. Therefore, the dielectric constant decreased. Furthermore, the dielectric constant of sample O4 was larger than that of W3. This is because a large number of impurities were introduced during the cooling process. The polarization was affected greatly by the impurities, and the dielectric constant increased.

From Figure 6, the loss factor order of different samples was as follows: A1 < A2 < O4 < W3. This is because of the impurities introduced during the cooling process. The relaxation polarization loss of impurities increased, so the loss factor increased. The loss of sample W3 was much greater than that of the other three samples. This is because a flat water film formed on the sample’s surface after the water cooling process, which increased the surface conductivity loss. Additionally, water molecules existed in the samples, and a dipole polarization loss was generated. Therefore, the dielectric loss increased greatly.

### 3.5. Test Results and Analysis of Space Charge Characteristics

The space charge distributions of different samples are shown in Figure 7.

From Figure 7, with the increase in field strength, the heteropolar space charge accumulations in all four samples increased to different degrees. Among them, the space charge density of sample O4 was the smallest. From the space charge distribution of sample A1, under the 20 kV/mm field strength, hetero-charges existed around the negative pole. This was because the impurity resolved at a high field strength. These were captured by the traps during the mobility process, and the hetero-space charges formed. Under the 40 kV/mm field strength, there were massively homopolar charges around the electrode. These charges were injected by the electrode. From the space charge distribution of sample A2, under the 20 kV/mm field strength, there were few hetero-charges around the negative pole. This is because air cooling changed the crystalline morphology of LDPE. The grain size decreased, and the traps were shallow. Thus, the space charges were easy to transfer, and the space charge accumulation decreased. However, under the 40 kV/mm field strength, more charges were injected by the electrode. This was because air cooling changed the surface morphology of LDPE, which caused work function changes between the electrode and the samples’ surface. From the space charge distribution of sample W3, under the 20 kV/mm field strength, there were no hetero-charges around the negative pole. Under the 40 kV/mm field strength, very small amounts of charges were injected by the positive electrode. The charge quantities in sample W3 were larger than in samples A1 and A2. From the space charge distribution of sample O4, under the 20 kV/mm field strength, the hetero-charges around the negative pole and charge in sample O4 were both fewer. The charges injected by the electrode were obviously restrained. This is because the crystalline morphology of LDPE was greatly affected by the processes of water cooling and oil cooling. The grain sizes decreased further, and the traps were shallower, so the space charges transferred more easily. Additionally, the changes in crystalline morphology also affected the samples’ surface morphology. The inhibiting effect of electrode injection was more visible.

The space charge distribution of four samples during the short-circuit moments of 10 s, 900 s and 1800 s are shown in Figure 8.

From Figure 8, with the prolongation of short-circuit time, the space charges accumulating in all samples decreased. From the space charge attenuation of sample A1 in short-circuiting, there were more remaining space charges at 10 s. Around the negative electrode, there were lots of hetero-charges. Around the positive electrode, there were lots of homopolar charges. A large number of negative charges were found in sample A1. At 1800 s, there were still a small number of remaining space charges. Both the hetero charges around the negative electrode and homopolar charges around the positive electrode decreased. Besides, the quantity of negative charges in sample A1 decreased. This is because the traps in sample A1 were deeper and the traps’ density was high. At the same time, more charges were injected by the electrode, so there were more remaining space charges in sample A1. However, with the prolongation of short-circuit time, the space charges captured by shallow traps could get out, which decreased the quantity of space charges. From the space charge attenuation of sample A2 in short-circuiting, the space charge distribution was different from that of sample A1. The homopolar charges existed in the negative electrode. There were many positive charges in sample A2. At 10 s, the charge peak around the negative electrode decreased, and the charge peak around the positive electrode increased. At the same time, the charge peak in sample A2 decreased. At 1800 s, the charges peak around the negative electrode and the positive electrode both increased, but the charge peak in sample A2 decreased. This illustrated that rapid air cooling changed the crystalline morphology of LDPE, and the attenuation characteristics of space charges were also affected [29]. From the space charge attenuation of samples W3 and O4 in short-circuiting, the space charge distributions of samples W3 and O4 were identical with those of sample A1. Whether at 10 s or 1800 s, the charge peak around the negative electrode and positive electrode both decreased. Besides, the charge peak in samples W3 and O4 also decreased. This illustrated that the space charges in samples W3 and O4 were seriously attenuated, and the quantities of the remaining space charges were lower. In summary, the remaining space charges quantities, in order of the four samples, was as follows: O4 < W3 < A2 < A1. This is because the water cooling and oil cooling further reduced the grain size. The trap depth became shallower, so the space charge transitions were easy to achieve.

## 4. Conclusions

For the purpose of changing the crystalline structure of the polymer matrix, in this article, different cooling media and cooling rates were used. Four LDPE samples were prepared with natural air cooling, rapid air cooling, water cooling and oil cooling. According to PLM and DSC tests, the effects of different cooling methods on polymers’ micro-crystalline morphologies could be explored. According to conductivity, dielectric frequency spectra and space charge tests, the influence mechanisms of micro-crystalline morphology on polymers’ dielectric properties were explored. Specific conclusions are as follows:From the PLM and DSC test results, the polymers’ crystalline morphology was greatly affected by cooling medium and cooling rate. The grain size order of the four samples was as follows: O4 < W3 < A2 < A1. At the same time, in sample O4, the grains dispersed uniformly. Besides, sample O4 possessed a fast crystallization rate, high crystallization temperature and perfect crystalline morphology.From the macroscopic test results, under a low electric field, the four samples’ conductivity increased with rising field strength. Under a field strength of 50 kV/mm, the conductivity order of the four samples was as follows: O4 < W3 < A2 < A1. Additionally, with different field strengths applied for 20min, the space charge density in sample O4 was the least. After short-circuiting for 30min, the remaining space charge quantity in sample O4 was also the least.Sample O4 possessed a complete crystalline structure. In this material, the crystal size was small, and the crystallization rate was high. The space charge effect was obviously restrained. Under a high electric field, the conductivity was low. Therefore, crystalline structure changes could improve polymers’ dielectric properties. This provides a theoretical foundation for follow-up research about polymer composite insulating materials.

## Figures and Tables

**Figure 1 materials-13-02791-f001:**
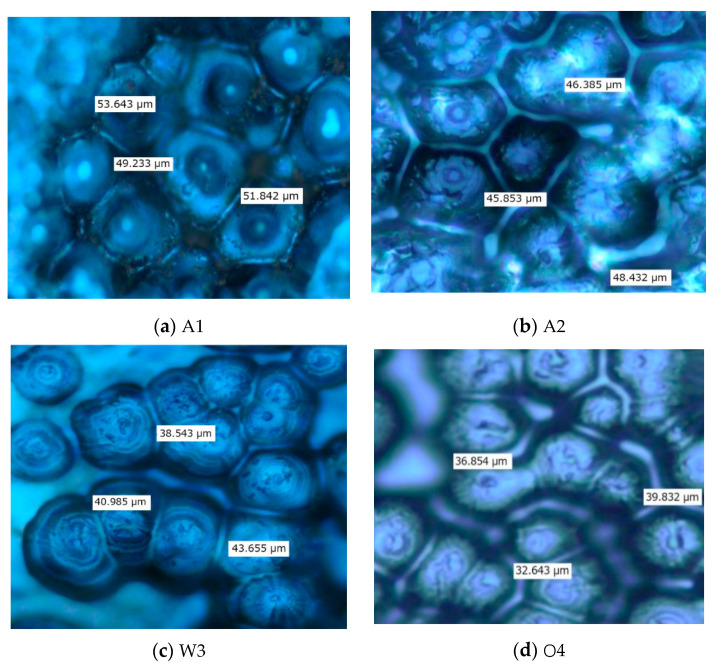
PLM pictures of LDPE.

**Figure 2 materials-13-02791-f002:**
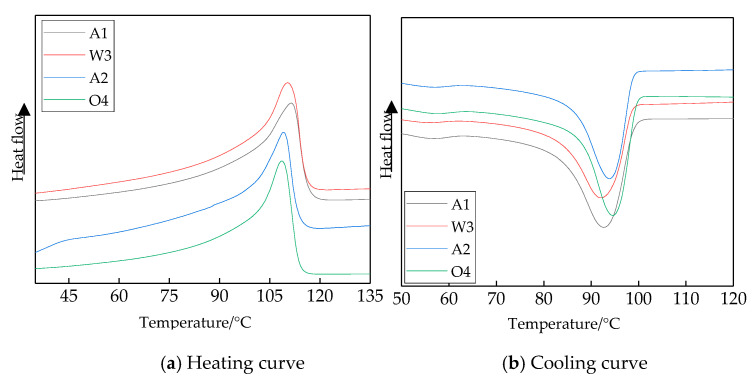
Heating and cooling curves of LDPE composites.

**Figure 3 materials-13-02791-f003:**
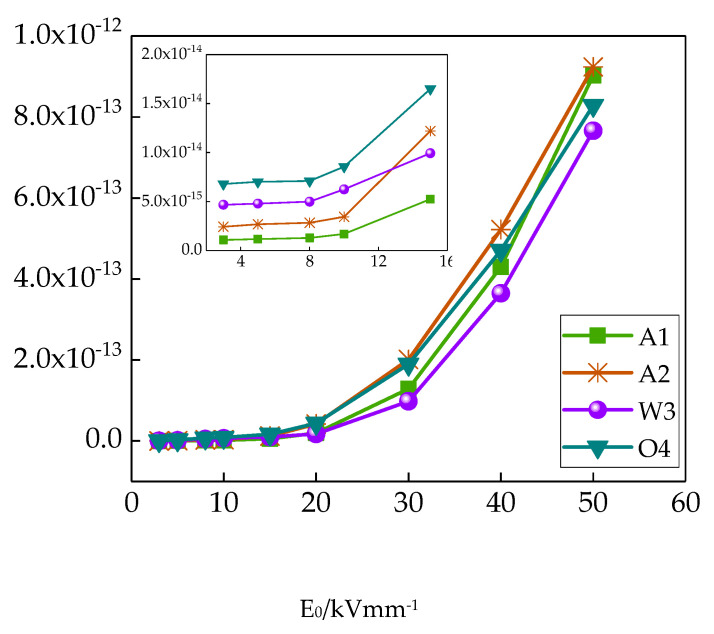
Curves of different LDPE samples’ conductivity changing with field strength.

**Figure 4 materials-13-02791-f004:**
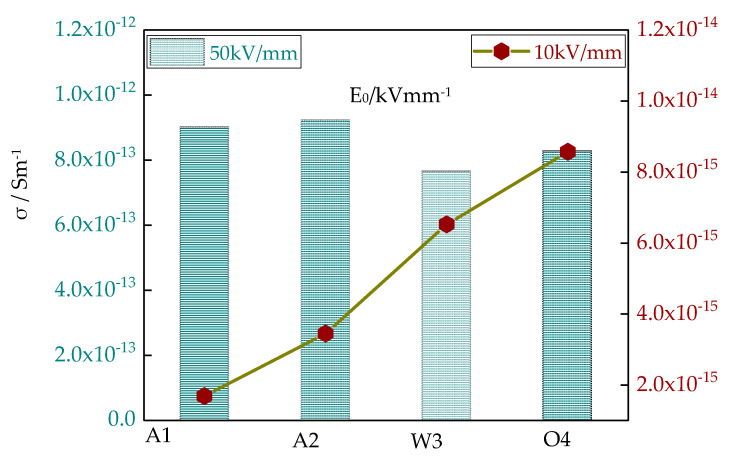
The conductivity line chart and bar chart of different cooling samples under 10 kV/mm and 50 kV/mm electric fields.

**Figure 5 materials-13-02791-f005:**
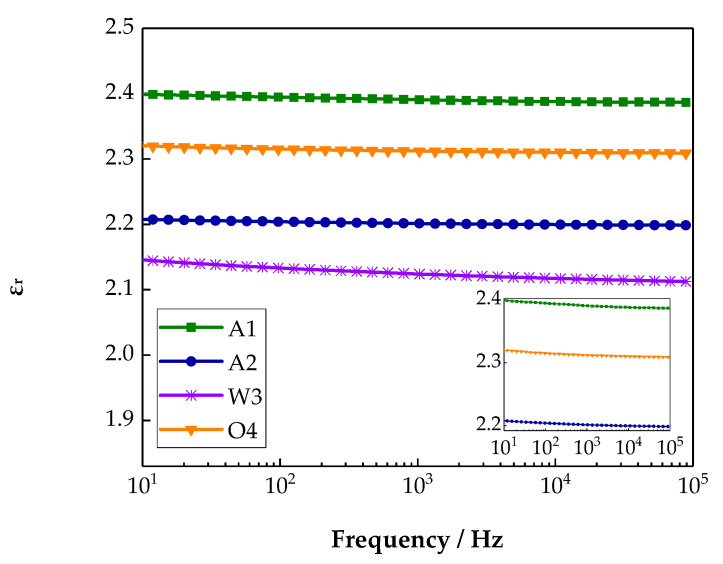
Frequency dependences of the relative permittivity of various samples.

**Figure 6 materials-13-02791-f006:**
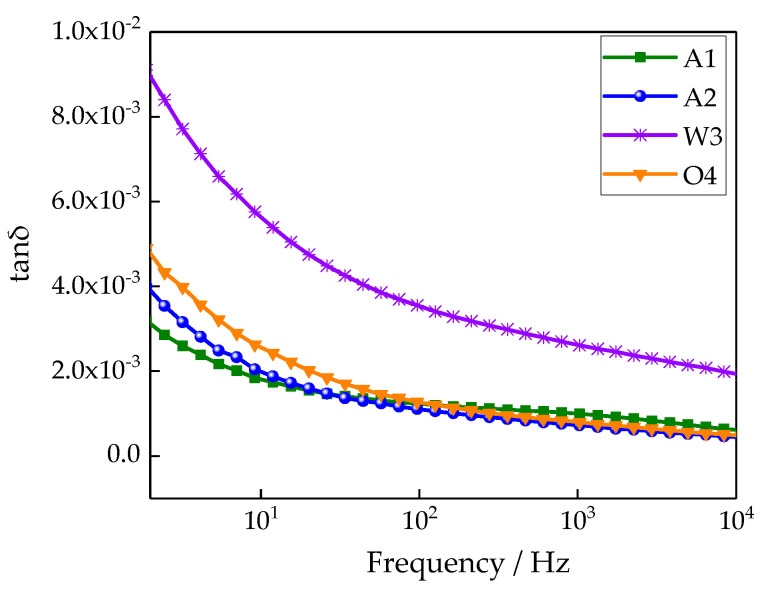
Frequency dependences of the dissipation factors of various samples.

**Figure 7 materials-13-02791-f007:**
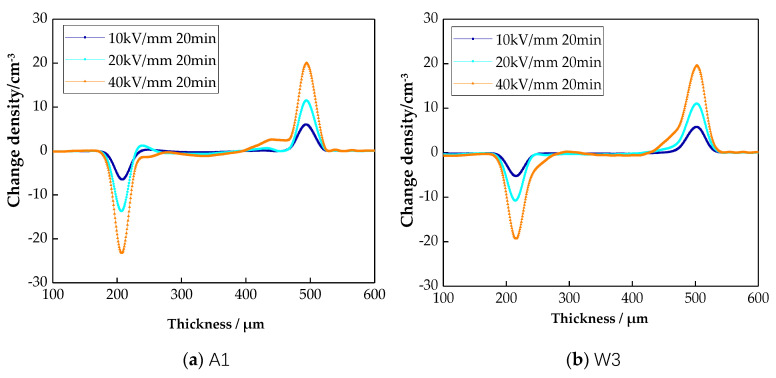
Space charge accumulations of LDPE under an electric field and different cooling media.

**Figure 8 materials-13-02791-f008:**
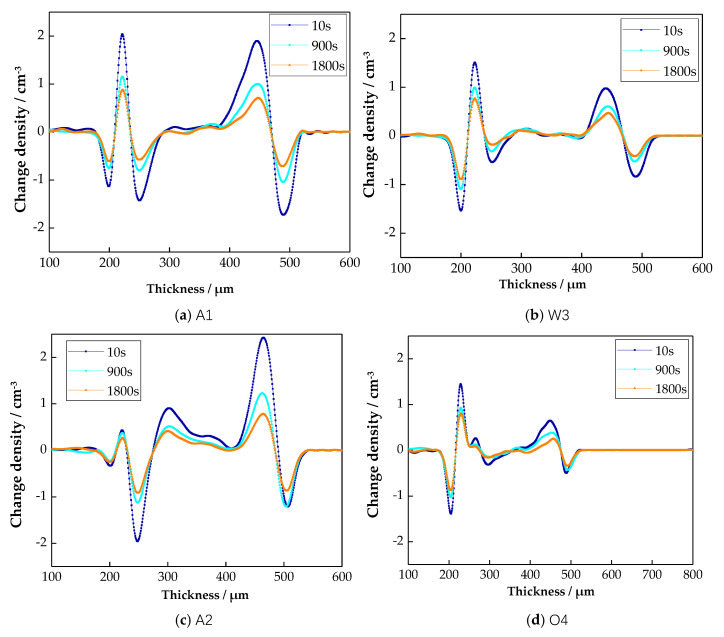
Space charge attenuation of LDPE under different cooling media when short-circuited.

**Table 1 materials-13-02791-t001:** Parameters of isothermal crystallization and melting processes.

Samples	*T*_c_ (°C)	Δ*T*_c_ (°C)	*T*_m_ (°C)	*X*_c_ (%)
A1	92.16	8.63	111.51	35.42
A2	92.66	8.12	109.27	36.8
W3	93.83	7.85	110.32	36.3
O4	94.50	7.43	108.76	37.2

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
