# Peer review of "The Effect of the Cooling Process on the Crystalline Morphology and Dielectric Properties of Polythene"

_materials, 2020, doi:10.3390/ma13122791_

Round 1

Reviewer 1 Report

The paper entitled ‘The effect of cooling process on crystalline morphology and dielectric properties of polythene’ considered for publication in the Materials presents a study of how the different cooling process influence the crystalline morphology and in crystallization process along with a study of its electrical response. LDPE is very attractive insulating material with great applicability, due its advanced mechanical and electrical properties, thus its investigation exhibits a lot of interest. In addition, some dielectric results are also cited along with the space charge distribution tests. The electric study was adequate and reveals good results, however the main purpose of this work is the relation of the grain size and crystallization with the electric response. There is a lack of morphological and structural analysis which needs to be performed in order to be able to correlate the electrical response with the material structure. I recommended this paper for publication after major revision.

In detail my comments can be seen below:

  1. In abstract there are acronyms that are not explained even in the text. What acronyms stand for should be written in the abstract and the first time they feature in the text.
  2. The introduction part should be enriched more with the novelty of the research and potential applicability (already cited in l.46-53) but it needs to be extended and by improving the literature references.
  3. Is there any orientation effect in the pressing process?
  4. Reference for the equation (1), and it would be beneficial a schematic representation of the 3-electrode system. Since you have measured in broadband dielectric analyzer the values of conductivity are also revealed. Thus the presentation of these values along with the ones that is already cited could be better and provide a better approach in both AC and DC conditions.
  5. Since the authors supports (and I agree) that the different cooling methods provides different grain size in each case. In this point I think it is necessary to perform also a size estimation distribution in every case. It seems from the PLM images that water and oil cooling results in smaller grain size, but it would be better to provide some results. Different microscopic technique would also work (SEM or TEM). In addition, in this part, DSC analysis must be also performed since the crystallization process is also affected. DSC analysis will provide a lot of valuable information regarding the crystallization in every case (crystallization temperature, crystallization percent etc.).
  6. In figure 2 conductivity parameter should be denoted by σ (Eq.1) not γ.
  7. Results in dielectric loss spectra should be in log x axis.
  8. The crystalline morphology affects the dielectric response of the system, this should be elaborate more precisely. A table with the crystallinity values along with conductivity and permittivity values will also be beneficial for the study.
  9. Conclusions should be improved, and present also the accomplishment of this work and how is this related to the novelty.
  10. The quality of English needs improvement.

Reviewer 2 Report

It is generally known, that the cooling rate influences the amount of crystallinity, when slow cooling provides time for greater amounts of crystallization to occur. Fast rates, on the other hand, such as rapid quenches, yield highly amorphous materials. Here the method of cooling is reported when the cooling speed and also different cooling procedure (air, oil, water) are used.

Please try to avoid to use of abbreviations in abstract before earlier specification (PLM, DSC) even if there are familiar.

Line 13 – “According to the PLM and DSC test of these four LDPE samples, the effect of different 13 cooling processes on polythene crystalline morphology could be studied.” – results of DSC measurements fully missing

Line 89; 92 - Stabilization of samples before measurement (especially water and oil cooling) should be clearly defined (how long, what temperature). As mentioned at line 207 – 209 – the problem could be connected with “contamination” of the sample surface. Is any chance to clean the surface before measurement?

Quality of figure 1 – not a sharp figure. Please include abbreviations (A1, A2, W3, O4) which is used it paper text also below figure (you have only text air natural cooling etc.).  Please use the same label in paper text and figures (all document). Now authors use abbreviations in text and fully name in figures.

Chapter 3.2 especially figures 2 and 3 – missing error bars – how many samples were tested in each method. Especially for the comparison on Fig. 3 at 50 kV/mm, are really differences caused by different cooling, or it is caused by variability defined by standard deviation?

Fig. 3 - Please specify clearly which y-axis represent which voltage (left 50 kV/mm, right 10 kV/mm?)

Line 189 - From figure 5, with the increase of frequency, the relative dielectric constant of different samples all decreased at different degrees.  – the decreasing is visible only for the sample water. To improve drop visibility, please include secondary horizontal grid.

Fig. 7, 8 - describe the axes including units

Important - for all method missing experiment measurement description – for example, number of sampling, used apparatus, experimental conditions, etc.

Conclusion – there are listed facts for single method results, but missing some complex summary or recommendation  

Round 2

Reviewer 1 Report

The authors addressed most of the relevant points previously indicated. I recommend publication after minor revision, because there are two additional points that should be also addressed in my opinion.

  1. There is no indication of the three-electrode system as it is mentioned in authors’ reply.
  2. Size estimation distribution in each different case would provide additional insight in the research.

Reviewer 2 Report

Line 13-14: Differential scanning calorimeter is a device, method is called Differential scanning calorimetry. At the same time, polarization microscope is a device not a method (microscopy).

 Formula 1: The naming of quantities in the formula is missing. Not for all readers could be familiar using of enthalpy of melting/crystallization. Also weight of the sample is important parameter for tested samples. Please include also this information (sample weight, not only thickness). Was the weight of teste samples identical?

As known, heat/cool/heat measuring cyclces are used for the DSC analysis. The authors correctly used the results from the first heating to analyse process of cooling.   

Line 188: Figure 2: Missing direction of endo/exo reaction. It is commonly used, that marker with direction is included in graph (endo up/down, exo up/down) – suppose, that Figure 2 is endo up (exo down).

Please use the same label in paper text and figures (in all document). Authors corrected it in text above but not in newly included figure 2 or Figure 5 where full name is used. Please use either abbreviations or full names, but the same throughout the document.

Response 5: In conductivity test, 10 group data per samples were tested, then averaged the results. Moreover, our team have made the similar study about the effect of cooling methods on polymers conductivity characteristics. Two results were high consistency.

I still think that error bars should be added to the graph. Even with regard to the fact that 10 samples were measured and the values averaged.

Line 362, just a trifle: authors used text “O4
